Seasonal dispersal and longitudinal migration in the Relict Gull Larus relictus across the Inner-Mongolian Plateau

Liu Dongping 1
Zhang Guogang zm7672@caf.ac.cn 1
Jiang Hongxing 1
Chen Lixia 1
Meng Derong 2
Lu Jun 1
1 Key Laboratory of Forest Protection of State Forestry Administration, Research Institute of Forest Ecology and Environment Protection, Chinese Academy of Forestry , Beijing , China
2 Cangzhou Normal University , Cangzhou , China
Pimm Stuart
Electronic publication date: 2017 May 25
Publication date: 2017
Volume: 5
Electronic Location ID: e3380
Received 2017 Mar 18; Accepted 2017 May 5
Copyright: ©2017 Liu et al.
Copyright year: 2017
Copyright holder: Liu et al.
License: This is an open access article distributed under the terms of the Creative Commons Attribution License, which permits unrestricted use, distribution, reproduction and adaptation in any medium and for any purpose provided that it is properly attributed. For attribution, the original author(s), title, publication source (PeerJ) and either DOI or URL of the article must be cited.
License URL: https://creativecommons.org/licenses/by/4.0/

Keywords: Relict Gull, Larus relictus, Satellite tracking, Longitudinal migration, Loop migration, Migration flexibility, Hongjian Nur

Funding: National Key Research and Development Program of China 2016YFC1201601 Wildlife Surveillance Program from State Forestry Administration of China This study was financially supported by the National Key Research and Development Program of China (2016YFC1201601) and Wildlife Surveillance Program from the State Forestry Administration of China. The funders had no role in study design, data collection and analysis, decision to publish, or preparation of the manuscript.

==============================
The Relict Gull Larus relictus is a globally vulnerable species and one of the least known birds, so understanding its seasonal movements and migration will facilitate the development of effective conservation plans for its protection. We repeatedly satellite-tracked 11 adult Relict Gulls from the Ordos sub-population in Hongjian Nur, China, over 33 migration seasons and conducted extensive ground surveys. Relict Gulls traveled ∼800 km between Hongjian Nur in northern China to the coast of eastern China in a predominantly longitudinal migration, following a clockwise loop migration pattern. The gulls migrated faster in spring (4 ± 2 d) than in autumn (15 ± 13 d) due to a time-minimization strategy for breeding, and they showed considerable between-individual variation in the timing of the autumn migration, probably due to differences in the timing of breeding. Gulls that made at least two round trips exhibited high flexibility in spring migration timing, suggesting a stronger influence of local environment conditions over endogenous controls. There was also high route flexibility among different years, probably due to variations in meteorological or habitat conditions at stopover sites. Relict Gulls stayed for a remarkably long time (234 ± 17 d) on their major wintering grounds in Bohai Bay and Laizhou Bay, between which there were notable dispersals. Pre-breeding dispersals away from the breeding area were distinct, which seemed to be a strategy to cope with the degradation of breeding habitat at Hongjian Nur. Overwhelming lake shrinkage on the breeding ground and at stopover sites and loss of intertidal flats on the wintering grounds are regarded as the main threats to Relict Gulls. It is crucial to make protection administrations aware of the great significance of key sites along migration routes and to promote the establishment of protected areas in these regions.

Introduction

The Relict Gull (Larus relictus) breeds at scattered sites on arid lakes in eastern Kazakhstan, Mongolia and the Russian Far East, but most breeding colonies, known as the Ordos sub-population, occur in Shaanxi and Inner Mongolia in northern China (He et al., 2002; BirdLife International, 2016). The global population of the Relict Gull has been estimated at 10,000–19,999 but has fluctuated considerably due to human disturbance and lake degradation at breeding sites as well as habitat loss on the wintering grounds in East Asia (He et al., 2005; BirdLife International, 2016). Therefore, the Relict Gull is recognized as vulnerable on the IUCN Red List (BirdLife International, 2016) and is listed among the First Class State Protected Wildlife in China.

Confirmed as a valid species as recently as 1971 (Auezov, 1971), the Relict Gull is one of the least known birds. Most previous research has focused on its breeding colonies, and only occasional winter records have been documented in Japan, Korea, Vietnam, eastern China and Hong Kong (Duff, Bakewell & Williams, 1991; He et al., 2002; Olsen & Larsson, 2004). Since the late 1990s, legflag marking and resighting has indicated that the Bohai Bay of China is an important wintering ground for the Ordos sub-population (He et al., 2002), and later field surveys have confirmed that Bohai Bay supports a considerable wintering population (Liu et al., 2006). However, the detailed migratory routes, stopover sites and wintering range of the Relict Gull remain poorly understood (Liu et al., 2006; BirdLife International, 2016).

Because most Relict Gulls breed at Hongjian Nur in northern China (Xiao et al., 2008) and winter on Bohai Bay, the migration routes of the species probably cover vast areas of the Inner-Mongolian Plateau, where several arid lakes may serve as stopover sites. Over the past several decades, the Inner-Mongolian Plateau has experienced increasingly significant lake shrinkage resulting from climate change and the increasing exploitation of underground minerals and groundwater resources (Tao et al., 2015). On the coast of eastern China, the potential wintering grounds of the Relict Gull, ongoing tidal land reclamation has had continuous, serious impacts on waterbirds (Yang et al., 2011; Ma et al., 2014), so the species may face serious threats from habitat loss along its migration routes. Identifying key stopover sites and determining the wintering range of the Relict Gull is critical for the conservation of this vulnerable species and would provide basic information to improve our understanding of the impacts of habitat loss on bird migration strategies.

We deployed Platform Transmitter Terminals (PTTs) on Relict Gulls breeding in Hongjian Nur in northern China to determine their migration routes, stopover sites, wintering grounds and seasonal dispersal. Based on satellite tracking data, we compared between- and within-individual variation in migration timing and routes and examined repeat autumn and spring journeys to analyze flexibility in individual migration behavior. We also conducted extensive ground surveys along the migration routes of the species to better understand the status of the Relict Gull population and its habitat at major stopover sites and on its wintering grounds.

Methods

Study site

Hongjian Nur (38°13′–39°27′N, 109°42′–110°54′E) is located at the junction of the Mu Us Desert and the Ordos Plateau, approximately 60 km south of the city Ordos in Inner Mongolia, China, at an elevation of 1,200 m. It used to be the largest desert freshwater lake in China, but the total area of the lake has been shrinking, from 55 km2 in 1997 to 32 km2 in 2013, due to drought and overexploitation of the groundwater. Hongjian Nur is an important breeding ground and stopover site for many waterbird species (Xiao et al., 2010), and it has been protected as a county-level nature reserve since 1997. Breeding Relict Gulls were first recorded on the islands of Hongjian Nur in 2000, and the breeding population has fluctuated considerably due to habitat changes in recent years, with the number of nests reaching a maximum of ca. 7700 in 2011 (Miao, 2014). In this study, Relict Gulls were captured on the largest nesting island (39°8.3′N, 109°52.2′E).

Gull capture and transmitter attachment

We captured 11 adult Relict Gulls (two in 2007, five in 2008, and four in 2010) on their nesting island using monofilament leg nooses; all capture attempts were made after the chicks had fledged to avoid disturbing the breeding activities. Upon capture, the gulls were placed in individual cloth bags and promptly processed, during which we recorded the mass, wing length and tarsus length of each gull.

The gulls were tagged with 9-g solar-powered PTTs (PTT-100; Microwave Telemetry, Inc., Columbia, MD, USA) in 2007 and with 12-g solar-powered PTTs (Model 12 GS; Northstar Science and Technology, LLC, VA, USA) in 2008 and 2010, using a Teflon ribbon back harness; the combined weight of the PTT and harness was 1.6–2.6% of the gull’s body mass. The gulls were also marked with orange Darvic-type leg flags. All gulls were released near their capture locations as soon as possible after processing.

Satellite telemetry locations and spatial analysis

The PTTs were tracked using the following transmitting cycles: 10 h on and 47 h off in 2007, 8 h on and 15 h off in 2008, and 8 h on and 23 h off 2010, and the Doppler-derived PTT location information was received by the CLS/Service Argos satellite tracking system (Argos, 2007). Location classes (LC) ranged from zero to three, which reflected location accuracy (Argos, 2007), and 1-sigma error radii of >1,000 m, 350–1,000 m, 150–350 m, and 150 m were reported for LCs 0, 1, 2 and 3, respectively. Auxiliary LC A, B and Z were not assigned accuracy estimates. We used the Douglas Argos-Filter Algorithm (v. 8.50) to identify and remove implausible auxiliary Doppler locations based on the distance moved, movement rate, and turning angle (Douglas et al., 2012).

Because the PTTs did not transmit continuously, we used the median date between the last point at the previous location and the first point at the new location to calculate timing. If the gap between dates was greater than ten days due to no data or LCs A, B, and Z data received from ARGOS, then timing was not calculated.

We used ArcView GIS (version 3.2; Environmental Systems Research, Redlands, CA, USA) to plot the telemetry locations and delineate the migratory routes. Stopover sites were defined as areas where birds moved less than 20 km during at least a 24-h period, and we used the X-tool extension to calculate the lengths of the migration segments between two adjacent stopover sites. To determine the wintering range of the Relict Gulls, we pooled the wintering locations and calculated 90% and 50% fixed kernel home ranges using a fixed kernel method with least squares cross validation for the smoothing factor (Worton, 1989) in the Animal Movement extension (Hooge & Eichenlaub, 1997).

Ground surveys

Google Earth 7.1.5.1557 (Google, Mountain View, CA, USA) was used to plot stopover sites and wintering locations and to identify potentially important areas based on known habitat requirements. Stopover sites that were heavily used by the tracked gulls were investigated at the time of year when they were most active, and we counted the number of Relict Gulls with the aid of 8 × 42 binoculars and 20–60 × 80-mm spotting scopes and recorded human disturbance and lake shrinkage (Table S1). To monitor the population dynamics on the wintering grounds, we conducted monthly surveys at 37 and four wintering locations (Fig. S1), as revealed by satellite tracking, on the Bohai Bay and Laizhou Bay, respectively, from September 2011 to April 2012. During each survey, two teams consisting of two to three members each simultaneously surveyed the wintering locations on Bohai Bay and Laizhou Bay. Counters arrived at specific locations two hours before low tide, and each performed separate counts from the shore on foot. Furthermore, the major habitat threats at each wintering location were recorded (Table S2).

Individual migration variation and data analysis

With individual as a factor in ANOVA, we compared between- and within-individual variation in migration timing and route (Vardanis et al., 2011; Stanley et al., 2012). Based on the satellite tracking results, Relict Gull migration was predominantly longitudinal, and the routes were relatively short; 114°E marked the midline of the routes (see ‘Results’) and was a suitable reference to determine the spatial variation in migration. Therefore, we examined four variables: departure date, latitude crossing 114°E, arrival date, and migration duration. For gulls with at least two round trips, we also calculated the repeatability (intra-individual correlation coefficient; Lessells & Boag, 1987) of migration timing and route.

We used t-tests to examine the within-route and autumn and spring differences in migration segments, and we used paired t-tests to compare the individual autumn and spring migration distances. In cases where individuals were tracked for more than one round trip, we used the mean values of the repeated seasons for each of these individuals in the comparison.

All statistical analyses were performed using SPSS software (version 22.0, IBM 2013). Results were given as mean ± SD with a significance level of 0.05 based on two-tailed tests.

Ethical note

All data collected as part of this study were approved by National Bird Banding Center of China (No. NBBC20070512). Field work was approved by State Forestry Administration (No. 33 Forestry Protection [2002]).

Results

In 2007, the PTT signals of two of the 11 tracked Relict Gulls (G1 and G2) were lost before the onset of autumn migration. For the remaining nine individuals (G3–G11), we obtained data of 20 complete autumn migration journeys and 13 complete spring migration journeys, including one individual with one round trip, two individuals with two round trips and two individuals with four round trips (Table 1). The complete dataset for these nine individuals contained 9,151 locations; 40.8% were high quality (LC 1–3). The average number of locations recorded per individual was 1,017 ± 976 (n = 9), and the average number of locations per day per individual was 1.7 ± 0.7 (n = 9, Table S3).

Table 1 Migration timing, number of stopover sites and migration distance of satellite-tracked Relict Gulls between 2008 and 2012.

Individuals	Autumn migration	Spring migration	
	Departure  date	Arrival date	No. of  stopovers	Length of  migration  segments  (km)	Departure  date	Arrival date	No. of  stopovers	Length of  migration  segments  (km)	
G3	1 Aug 2008	3 Aug 2008	0	810	8 Apr 2009	15 Apr 2009	1	470/380	
	31 Jul 2009	3 Aug 2009	1	470/420	–	–	–	–	
G4	26 Jul 2008	1 Sep 2008	3	400/40/30/420	4 Apr 2009	6 Apr 2009	0	700	
	28 Jul 2009	27 Aug 2009	5	290/50/70/50/30/500	5 Apr 2010	9 Apr 2010	0	690	
	11 Aug 2010	11 Sep 2010	2	440/30/420	5 Apr 2011	12 Apr 2011	0	680	
	26 Jul 2011	6 Sep 2011	2	350/90/460	6 Apr 2012	7 Apr 2012	0	680	
G5	3 Aug 2008	19 Aug 2008	1	440/470	31 Mar 2009	6 Apr 2009	0	670	
	18 Jul 2009	11 Aug 2009	3	510/80/40/410	1 Apr 2010	3 Apr 2010	0	710	
	30 Jul 2010	14 Aug 2010	2	380/90/430	8 Apr 2011	11 Apr 2011	0	680	
	27 Jul 2011	28 Jul 2011	0	670	10 Apr 2012	14 Apr 2012	0	670	
	3 Aug 2012	17 Aug 2012	1	490/440	–	–	–	–	
G6	2 Aug 2008	12 Aug 2008	1	490/470	–	–	–	–	
G7	22 Jul 2008	3 Aug 2008	2	420/100/370	10 Apr 2009	13 Apr 2009	1	810/240	
	21 Jul 2009	22 Jul 2009	0	670	2 Apr 2010	7 Apr 2010	0	670	
G8	2 Sep 2010	24 Sep 2010	2	240/120/380	–	–	–	–	
G9	30 Jul 2010	14 Aug 2010	1	390/300	–	–	–	–	
G10	13 Aug 2010	14 Aug 2010	0	670	–	–	–	–	
G11	26 Jul 2010	5 Aug 2010	2	490/100/300	3 Apr 2011	5 Apr 2011	1	550/180	
	21 Jul 2011	23 Jul 2011	0	710	31 Mar 2012	3 Apr 2012	0	810	
	21 Jul 2012	23 Jul 2012	0	740	–	–	–	–	

Migration timing

During the autumn migration, Relict Gulls departed their breeding grounds on July 31st (range = July 18th–September 2nd, n = 20), with 80% of individuals leaving between July 21st and August 3rd. After migrating for 15 ± 13 days (range = 1–42 days, n = 20), gulls arrived at their wintering grounds on August 16th (range = July 22nd–September 24th, n = 20; Table 1) and remained for 234 ± 17 days (range = 206–255 days, n = 13), an extremely long duration. The gulls departed their wintering grounds on April 5th (range = March 31st–April 10th, n = 13) and arrived at their breeding grounds on April 9th (range = April 3rd–April 15th, n = 13) after migrating for 4 ± 2 days (range = 1–7 days, n = 13; Table 1).

Migration routes

Unlike that of most bird species, Relict Gull migration constituted a far greater shift in longitude than latitude (Δ longitude =9.4 ± 0.7°, Δ latitude =1.7 ± 0.5°, n = 29; Fig. 1). During the autumn migration, most gulls (six of the nine gulls in 13 of 20 autumn migration seasons) took inverted V-shaped routes; after departing from Hongjian Nur, they flew northeast across the Inner-Mongolian Plateau and rested at wetland complexes formed by Chahan Nur, Xiyan Nur, and Anguli Nur, etc., at the border between Inner Mongolia and Hebei Province (Fig. 1). After refueling at these wetland complexes, the gulls headed southeast and finally arrived at their wintering grounds on the coast of eastern China. In the remaining seven seasons, the gulls took quite direct routes across Shanxi and Hebei Province.

Figure 1 Relict Gull migration routes and stopover sites determined by satellite tracking during autumn (red lines) and spring (green lines) journeys from 2008–2012.

1-Wulanhushaohaizi & Baiyin Nur, 2-Hanhaizi, 3-Chahan Nur, 4-Kangbo Nur, 5-Xiyan Nur, 6-Anguli Nur, 7-Huanggai Nur & Sangai Nur, 8-Yanghe Reservoir, 9-Huangqihai, 10-Daihai, 11-Dongyulin Reservoir, 12-Cetian Reservoir, 13-Xiahewan Reservoir, 14-Dangyangqiao Reservoir, 15-Shuangrushan Reservoir.

During the spring migration, gulls used similar routes as in autumn, but most (in 11 of the 13 seasons) took the more direct routes. Thus, Relict Gulls generally followed a clockwise migration loop (Fig. 1).

Variation in and repeatability of migration timing and routes

Individual had a significant effect on all timing variables in autumn (Table 2), but there was no effect on the timing of the spring migration. Tests of within-individual repeatability indicated that migration variables were more repeatable in autumn than in spring and in timing than in route (Table 3). High repeatability (One-way ANOVA, F3,10 = 17.42, p < 0.001, r = 0.80) was found in the winter arrival date, for which individuals differed between years by an average of ±8 d, and the duration of the autumn migration was also highly repeatable (One-way ANOVA, F3,10 = 14.28, p = 0.01, r = 0.76), with an average difference of ±7 d between years.

Table 2 One-way ANOVA results and p-values of the effects of an individual on migration timing and route variables for nine Relict Gulls.

Variable	df	F	p-value	
Autumn migration	
Autumn departure date	8, 11	6.20	0.04*	
Latitude crossing 114°E	8, 11	1.51	0.26	
Winter arrival date	8, 11	11.76	0.000***	
Autumn migration duration	8, 11	7.72	0.01*	
Spring migration	
Spring departure date	4, 8	1.24	0.37	
Latitude crossing 114°E	4, 8	3.70	0.06	
Breeding arrival date	4, 8	1.53	0.28	
Spring migration duration	4, 8	0.87	0.52	
Notes.

* p < 0.05.

*** p < 0.001.

Table 3 Repeatability (r) of migration timing and route variables for four Relict Gulls with at least two round trips.

Variable	df	F	r	p-value	
Autumn migration	
Autumn departure date	3, 10	1.60	0.13	0.25	
Latitude crossing 114°E	3, 10	1.79	0.17	0.20	
Winter arrival date	3, 10	17.42	0.80	0.000***	
Autumn migration duration	3, 10	14.28	0.76	0.01*	
Spring migration	
Spring departure date	3, 8	0.56	−0.18	0.66	
Latitude crossing 114°E	3, 8	0.98	0.13	0.27	
Breeding arrival date	3, 8	0.98	−0.01	0.45	
Spring migration duration	3, 8	0.23	−0.37	0.88	
Notes.

* p < 0.05.

*** p < 0.001.

Stopover sites and distances

After departing from Hongjian Nur, the gulls traveled 839 ± 116 km (n = 20, range = 670–1,040 km) and made a diverse number of stopovers (average =1.4, range = 0–5); in 17 of the 20 autumn migration journeys, all nine tracked gulls made two or fewer stopovers before reaching their wintering grounds. The gulls traveled 350 ± 216 km (range = 30–810 km, n = 48) per migration segment (Table 1), but in the journeys with two or more stopover sites, the first and last migration segments (401 ± 73 km, n = 18) were significantly longer than the remaining segments (66 ± 31 km, n = 14; t-tests, t = 16.13, df = 30, p < 0.001), indicating that the stopover sites were concentrated in the middle portion of the route (Fig. 1). Gulls used the Chahan Nur, Xiyan Nur and Anguli Nur wetland complexes at the border between Inner Mongolia and Hebei Province, where the gulls stayed for a total of 151 days (66.2% of the total duration) in 14 of the 28 autumn migration journeys (Table 4).

Table 4 Migration stopover sites, stopover frequencies and accumulated days for tracked Relict Gulls breeding at Hongjian Nur in China between 2008 and 2012.

Stopover sites	Coordinates  (latitude, longitude)	Stopover frequency  (no. migration journeys)	Accumulated  duration  (no. days)	
Autumn migration	
Chahai Nur	41.462°N, 113.901°E	6	105	
Xiyan Nur	41.522°N, 114.221°E	4	26	
Anguli Nur	41.289°N, 114.431°E	4	20	
Cetian Reservoir	39.980°N, 113.677°E	1	14	
Xiahewan Reservoir	39.750°N, 114.300°E	1	13	
Kangbu Nur	41.273°N, 113.890°E	2	12	
Huanggai Nur & Sangai Nur	41.350°N, 114.726°E	3	11	
Wulanhushaohaizi & Baiyin Nur	41.524°N, 113.270°E	1	10	
Hanhaizi	41.462°N, 113.523°E	1	7	
Huangqihai	40.830°N, 113.280°E	2	6	
Daihai	40.583°N, 112.757°E	1	2	
Yanghe Reservoir	40.543°N, 115.106°E	1	1	
Dongyulin Reservoir	39.361°N, 112.623°E	1	1	
Spring migration	
Wulanhushaohaizi & Baiyin Nur	41.524°N, 113.270°E	1	6	
Shuangrushan Reservoir	38.538°N, 112.607°E	1	1	
Dangyangqiao Reservoir	40.036°N, 111.625°E	1	1	

In spring, the gulls traveled 738 ± 109 km (n = 13, range = 670–1,050 km), and most (four of the five gulls in 10 of the 13 spring migration journeys) migrated directly to their breeding ground at Hongjian Nur without a stopover (Table 1).

On an individual basis, there was no significant difference between the autumn and spring migration distances (paired t-test, t = 1.08, p = 0.34, n = 5 individuals), but the migration segments in spring (599 ± 187 km, n = 16) were significantly longer than those in autumn (t-test, t = 4.136, df = 62, p < 0.001).

Wintering grounds and wintering dispersal

The wintering grounds of the Relict Gulls stretched along the coast from Changli (39.480°N, 119.257°E) in Hebei Province south to Jimo (36.446°N, 120.798°E) in Shandong Province (Fig. 2). The gulls wintered almost exclusively on Bohai Bay and Laizhou Bay in the Bohai Sea, except G11, who stayed on the Bay of the Yellow Sea in January 2011. The Bohai Bay wintering ground ranged from Leting in Hebei Province south to Zhanhua in Shandong Province, and the Laizhou Bay wintering ground was from Kenli south to Shouguang in Shandong Province. The core area of the wintering home range covered the coast in Tianjin, Huanghua and Luannan in Hebei Province and Haixing, Wudi and Kenli in Shandong Province and was 1,380 km2 in size (50% fixed kernel, Fig. 2).

Figure 2 Wintering home range and wintering dispersal of satellite-tracked Relict Gulls on Bohai Bay and Laizhou Bay from 2008–2012.

The size of the arrows roughly indicates the number of dispersal gulls.

For all 20 autumn migration journeys, all nine gulls arrived at Bohai Bay first but then exhibited distinct dispersal patterns between Bohai Bay and Laizhou Bay in 12 of the 13 wintering seasons, although in one season, G4 wintered solely on Bohai Bay. The gulls migrated from Bohai Bay to Laizhou Bay on 28th December (range = December 1th–February 4th, n = 12), and in nine (75%) of the 12 seasons, the gulls returned to Bohai Bay on March 13th (range = February 16th–April 4th Fig. 2). In one season, G11 stayed at Laizhou Bay until departure for spring migration, and in the remaining two seasons, insufficient data prevented us from determining if the gulls returned to Bohai Bay. Furthermore, G11 exhibited complex round-trip dispersal in Bohai Bay, Laizhou Bay and the Coast of the Yellow Sea in the 2010–2011 wintering season.

A maximum of 11,800 Relict Gulls were recorded on Bohai Bay and Laizhou Bay in February during the 2011–2012 wintering season. Ground surveys also indicated that the number of wintering Relict Gulls on Laizhou Bay gradually increased each month and peaked at 6,500 in February, confirming the dispersal pattern between Bohai Bay and Laizhou Bay revealed by satellite tracking (Fig. 3).

Figure 3 Number of wintering Relict Gulls on Bohai Bay and Laizhou Bay by month from 2011–2012.

Pre-breeding dispersal

Of the five Relict Gulls back to breeding ground, four showed novel pre-breeding dispersal from Hongjian Nur to the west, north or northeast in nine of the 13 breeding seasons. Upon arriving at Hongjian Nur during the spring migration, Relict Gulls refueled there for 4 ± 1 days (range = 2–7 days, n = 9) and then traveled 204 ± 146 km (range = 100–540 km, n = 9) to the following areas: a wetland complex in Ordos (where three gulls stayed for 42 days in four seasons, one stayed for a whole breeding season), Yellow River stretch in Baotou (two gulls stayed for 36 days in three seasons), a wetland complex in Tuoketuo (one gull stayed for 20 days in one season) and Wulanhushaohaizi (one gull stayed for 13 days in one season, Fig. 4). The gulls (G4, G5 and G7) exhibited low fidelity to the pre-breeding dispersal areas from year to year. After dispersal for 17 ± 6 days (range = 7–27 days, n = 8), most of the Relict Gulls, except G11, returned to their breeding grounds at Hongjian Nur on April 30th (range = April 27th–May 3rd).

Figure 4 Pre-breeding dispersal sites and direction of satellite-tracked Relict Gulls after returning to Hongjian Nur during the spring migration from 2009–2012.

Discussion

Migration timing, routes and stopover sites

This report is the first description of the migration details of the Relict Gull, a vulnerable species that heavily relies on arid lakes in Asia. Coincident with banding recovery and field observation results (He et al., 2002; Liu et al., 2006), the adult Relict Gulls from the Ordos sub-population mainly migrate across the Inner-Mongolia Plateau and winter on the coast of Bohai Sea. The migration of the Relict Gull is notable for its extensive change in longitude relative to latitude, which was also documented in the California Gull Larus californicus and the Pacific Gull Larus pacificus (Woodbury & Knight, 1951; Pugesek, Diem & Cordes, 1999), and the species generally exhibits a clockwise loop migration, following relatively shorter and more southward routes in spring compared to autumn to avoid the influence of cold weather (Fig. S2) and to quickly return to their breeding grounds.

A significantly faster migration was revealed in spring (4 ± 2 d) than in autumn (15 ± 13 d), due to a time-minimization strategy for breeding (Alerstam, 2011). At the extremes in both spring and autumn, there were gulls (G4, G5, G7 and G10) that accomplished their migration in less than one day, with traveling speeds of more than 30 km/h. Considering that the different transmitting cycle for PTTs in different years resulted in different sampling efficiency and switched off PTTs may cause incomplete sampling especially over a short period, the migration duration might be overestimated. Compared with other Larus species (Zhang et al., 2014), Relict Gulls arrived at their wintering grounds quite early and stayed for extremely long durations, and we suggest this is a response to hostile habitat conditions experienced en route that have resulted from remarkable lake shrinkage on the Inner-Mongolian Plateau (Tao et al., 2015).

Migration variability and repeatability

Relict Gulls exhibited considerable between-individual variation in the timing of the autumn migration, which, although not directly evident from our dataset, might be explained by breeding timing and breeding success. A delayed breeding season (e.g., laying a second clutch after an initial failure) resulted in both delayed departure from the breeding grounds and delayed arrival to the wintering grounds (G8 in 2010, Table 1) due to short stopovers.

Repeated tracking of the same individuals during consecutive years can improve our understanding of spatial and temporal variability and the role of flexibility in animals’ migratory behavior (Vardanis et al., 2011). Contrary to most bird migration habits (Vardanis et al., 2011; Stanley et al., 2012; López-López, García-Ripollés & Urios, 2014), Relict Gulls did not show high repeatability in the timing of spring migration from year to year, which suggests a stronger influence of the local environment conditions than endogenous controls; inclement weather on Bohai Bay and the Inner-Mongolian Plateau may have a particularly profound impact on the onset and arrival date of spring migration (Fig. S2) However, gulls exhibited highly repeatable winter arrival dates, which might be explained by higher predictability of environmental conditions along the migration routes at the end of the summer compared to just before the onset of spring migration (López-López, García-Ripollés & Urios, 2014). There was a high degree of flexibility in the routes followed by the same individual in different years, probably due to variations in meteorological conditions (Mellone et al., 2011) or habitat conditions at stopover sites (Table S1).

Wintering and pre-breeding dispersal

Our results confirmed that Bohai Bay is important for wintering Relict Gulls (Liu et al., 2006), and we determined that Laizhou Bay is another major wintering ground. The wintering dispersal from Bohai Bay to Laizhou Bay indicates that Relict Gulls may move southward along the coast to avoid hostile weather conditions (Fig. S2) or to follow southward shifts in maritime currents and food sources. Although our tracked adult gulls wintered stably on the coast of the Bohai Sea, there were irregular wintering records further south in Jiangsu, Shanghai, and Hong Kong in China and even in Japan and Vietnam, and most of these individuals were juveniles (Su et al., 1998; He et al., 2002). Moreover, banding and recovery records indicate that juveniles may winter from Jinzhou (Liaodong Bay of Bohai Sea) in Liaoning Province southwest to Yimen in Yunnan Province and prolong their stay at Bohai Bay until May (Table S4). We suggest that juvenile Relict Gulls, as revealed in some other Larus species (Pugesek, Diem & Cordes, 1999; Marques et al., 2009), were distributed over a larger range because they had more time and thus less need to follow the migration route between the breeding and wintering grounds; therefore, juveniles had a greater opportunity to select breeding sites other than their natal site.

Pre-breeding dispersals away from the breeding area were not an individual characteristic since gulls (G4 and G11) that undertook dispersal one year did not necessarily do so in other years (Data S1). Moreover, gulls (G4, G5 and G7) used different dispersal areas from year to year (Data S1). Pre-breeding dispersal seems to be a strategy to cope with the degradation of breeding habitat at Hongjian Nur by seeking new breeding areas. This is well exemplified by G11 who bred at Hongjian Nur in 2011 and did not engage in pre-breeding dispersal to a distant site, but pursued pre-breeding dispersal west to Ordos and bred there in 2012. Due to lake shrinkage at Hongjian Nur, the number of Relict Gull breeding pairs continuously decreased from ca. 7,700 in 2011 to 4,000 in 2016, but increasing numbers of breeding pairs have recently been discovered in the surrounding wetlands (Ren & He, 2015). Of the six pre-breeding dispersal areas revealed in this study (Fig. 4), three (the wetland complex in Ordos, Yellow River stretch in Baotou, and Wuliangsuhai) have been confirmed to be breeding areas since 2009 (Miao, 2014; Ren & He, 2015).

Threats along migration routes

Ongoing lake shrinkage on the Inner-Mongolian Plateau (Tao et al., 2015) has already profoundly impacted Relict Gulls, whose breeding colonies have been shifting over the past three decades (He et al., 2005; Ren & He, 2015). Since the first discovery of Relict Gulls in 1987, the number of breeding pairs at Boerjiang Nur reached a peak of 3,587 in 2000 before dramatically decreasing until the lake was abandoned in 2004 due to lake dried up (Miao, 2014). On the current Hongjian Nur breeding grounds, the number of breeding pairs has decreased since 2011 due to habitat change of the breeding island resulting from lake shrinkage. Currently, Relict Gull breeding colonies tend to scatter and shift to the surrounding lakes (Ren & He, 2015). As a result, breeding pairs will definitely invest more in nest site competition. Moreover, our ground surveys indicated that most stopover sites for Relict Gulls have faced major human disturbances in the forms of tourism and grazing with varying degrees of fishing, pollution and dredging (Table S1). Lake shrinkage was overwhelming at the stopover sites; Anguli Nur, Huanggai Nur and Saigai Nur were nearly dried up (Table S1). Our results suggest that changes in the quality, quantity and spatial distribution of suitable stopover sites have a profound impact on the migration strategy of the Relict Gull.

Due to rapid economic growth in China, coastal wetlands have been continuously reclaimed during recent decades, causing serious impacts on waterbirds (Yang et al., 2011; Ma et al., 2014). On the wintering grounds of Relict Gulls, 95% of the surveyed Relict Gull wintering locations (n = 37) on Bohai Bay face significant threats. Intertidal mudflat reclamation for aquaculture and harbor and oilfield construction has been found at 70% of the locations (Table S2). As a result of habitat loss, it is common to find large numbers of Relict Gulls concentrated into a small area on their wintering grounds, which makes the population extremely vulnerable to threats; for example, a serious oil spill could devastate these birds.

Our results revealed the most important sites along the migration routes of Relict Gulls. It is very important to make the local government aware of the great significance of these key sites for Relict Gulls and to promote the establishment of protected areas in these regions. Furthermore, as the species relies heavily on vulnerable arid lakes, the Relict Gull provides an opportunity to understand the degree of phenotypic plasticity in migration in response to environmental change.

Supplemental Information

Table S1 Human disturbance and habitat condition at major stopover sites for Relict Gull

By checking water level change since 1980s (Wang & Dou, 1998 ∗), we roughly ranked lake shrinkage as (A) no shrinkage or slight shrinkage (+), water surface area decreased by less than 10%; (B) medium (+ +), water surface area decreased by 10%∼50%; (C) serious (+ +  +), water surface area decreased by more than 50%; or (D) dried up (+ +  +  +). ∗ Wang S, Dou H. 1998. Lakes of China. Beijing: Science Press.

Click here for additional data file.

Table S2 Threats and their scope and duration in wintering locations for Relict Gulls on Bohai Bay

Threats include (A) intertidal mudflat reclamation; (B) tourism; (C) collection of sea food; (D) mining; (E) wind power generation; and (F) no threats. According to the spatial proportion affected, scope of threats was rated as (A) regional, <5%; (B) scattered, 5–15%; (C) widespread, 15–50%; or (D) entire, >50%. Durability of threats was rated as (A) short-term, <5 years; (B) medium-term, 5–20 years; (C) long-term, 20–100 years; or (D) permanent, >100 years. Location sees Fig. S1.

Click here for additional data file.

Table S3 Performance of satellite transmitters for Relict Gulls breeding at Hongjian Nur in 2007, 2008 and 2010

Click here for additional data file.

Table S4 Recovery records of juvenile Relict Gulls ( ≤two-year-old) in non-breeding season

All of the recorded gulls were banded when they were chicks. The recovery records indicate that juveniles distribute in a larger wintering range relative to satellite tracked adults. Data were accessed from China National Bird Banding Database ( http://www.chinanbbc.net).

Click here for additional data file.

Figure S1 Wintering survey locations for Relict Gull on Bohai Bay (sites 1–37) and Laizhou Bay (sites 38–41) from September 2011 to April 2012

1-Luanhe River Estuary, Leting Co.; 2-Daqinghe River Estuary, Leting Co.; 3-Puti Island, Leting Co.; 4-Xidajian, Leting Co.; 5-Xiaoqinghe River Estuary, Leting Co.; 6-Shuohe River Estuary, Tanghai Co.; 7-Qinglonghe River Estuary, Tanghai Co.; 8-Zuidong, Luannan Co.; 9-Nanbao, Luannan Co.; 10-Nanbao Oil Field, Luannan Co.; 11-Beibao, Luannan Co.; 12-Heiyanzi, Fengnan Co.; 13-Dashentang Power Field, Hangu of Tianjin; 14-Dashentang Wharf, Hangu of Tianjin; 15-Dongjiang Power Field, Hangu of Tianjin; 16-Central Fishing Port, Hangu of Tianjin; 17-Caijiabao, Hangu of Tianjin; 18-Chengtougu, Tanggu of Tianjin; 19-Haibinyuchang, Tanggu of Tianjin; 20-Duliujian River Estuary, Tianjin; 21-Ziyaxinhe River Estuary, Tianjin; 22-Qikou, Huanghua; 23-Zhangjuhe River, Huanghua; 24-Houtangbao, Huanghua; 25-Nanpaihe River, Huanghua; 26-Fanjiabao, Huanghua; 27-Guanjiabao, Huanghua; 28-Xiaoxinbao, Huanghua; 29-Xujiabao, Huanghua; 30-Yangjiabao, Huanghua; 31-Fengjiabao, Huanghua Harbor; 32-East Beach of Zhongtiegongsi, Huanghua Harbor; 33-Zhangweixinhe River Estuary, Huanghua Harbor; 34-East Bay of Dahekou Island, Huanghua Harbor; 35-Majiahe River Estuary, Wudi Co.; 36-Binzhou Harbor; 37-Taoerhe River Estuary, Wudi Co.; 38-Zhimaihe River Estuary, Guangrao Co.; 39-Xiaoqinghe River Estuary, Shouguang Co.; 40-Weihe River Estuary, Changyi; 41-Jiaolaihe River Estuary, Changyi

Click here for additional data file.

Figure S2 Daily temperature during spring migration (A) and winter dispersal (B), extracted from four typical meteorological station s (C)

(A) Temperature difference between southward and northward routes during spring migration from 2009 to 2012. Daily temperature varied greatly (range = −8 13;) in northward route among years, which may have impact on the onset and arrival date of spring migration. There are considerable proportion of days with temperature below zero in northward routes, which may explain why Relict Gulls tended to follow warmer southward routes in spring migration. (B) Difference of temperature (average from 2008 to 2012) between Bohai and Laizhou Bay during winter dispersal. The two reference lines for X axis indicate average time period (28 Dec–13 Mar) for Relict Gulls to disperse from frozen Bohai Bay to warmer Laizhou Bay. Temperature data were extracted from four typical meteorological station s (C) to represent northward & southward routes and Bohai & Laizhou Bay, respectively. Data accessed from NOAA (available at https://gis.ncdc.noaa.gov/maps/ncei#app=clim&cfg=cdo&theme=hourly&layers=1&node=gis).

Click here for additional data file.

Data S1 Migration details on timing, stopovers and seasonal dispersals

Different pre-breeding areas are indicated by colors. red-Ordos wetlands, blue-Yellow River strech at Baotou, green-Tuoketuo, grey-Daihai Lake, orange-Wulanhushaohaizi, purple-Wuliangsuhai, white-whole breeding season at Ordos. ∗ indicates signal lost. ∗∗signals were commonly lost for several days due to inclement weather in winter.

Click here for additional data file.

Data S2 Longitudinal migration character

Red texts indicate incomplete wintering data which were not included in the analysis.

Click here for additional data file.

Data S3 Variables for between-individual and within-individual test in migration timing and route

Click here for additional data file.

Data S4 Wintering count of Relict Gulls on locations of Bohai Bay and Laizhou Bay

Locations see Fig. S1—indicates not surveyed.

Click here for additional data file.

We thank Zhongqiang Wang Caie Hu and Yongqi Ren for their assistance with the field work. We thank two anonymous reviewers for their valuable comments to improve the manuscript.

Additional Information and Declarations

Competing Interests

Author Contributions

Animal Ethics

Field Study Permissions

The authors declare there are no competing interests.

Dongping Liu conceived and designed the experiments, performed the experiments, analyzed the data, wrote the paper, prepared figures and/or tables, reviewed drafts of the paper.

Guogang Zhang conceived and designed the experiments, performed the experiments, contributed reagents/materials/analysis tools, reviewed drafts of the paper.

Hongxing Jiang, Lixia Chen and Derong Meng performed the experiments, reviewed drafts of the paper.

Jun Lu contributed reagents/materials/analysis tools, reviewed drafts of the paper.

The following information was supplied relating to ethical approvals (i.e., approving body and any reference numbers):

All data collected as part of this study were approved by the National Bird Banding Center of China.

The following information was supplied relating to field study approvals (i.e., approving body and any reference numbers):

Field work was approved by the State Forestry Administration.

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
