# Peer review of "Seasonal dispersal and longitudinal migration in the Relict Gull Larus relictus across the Inner-Mongolian Plateau"

_PeerJ, doi:10.7717/peerj.3380_

## Round 0.1 · original submission · Minor Revisions

As you can see, my three reviewers only had minor comments. I, too, thought this was a most interesting paper and very straightforward. Please address all their concerns and let us have your revision back promptly.

Reviewer 1 ·

Basic reporting

See attachement

Experimental design

See attachement

Validity of the findings

See attachement

Additional comments

See attachement

Annotated reviews are not available for download in order to protect the identity of reviewers who chose to remain anonymous.

Reviewer 2 ·

Basic reporting

.

Experimental design

.

Validity of the findings

.

Additional comments

I like the article. There was nothing in it that jumped out at me as erroneous or dubious. The migrations they have plotted fit with our understanding of Relict Gull through observations in Beijing and on the coast at Tianjin and Hebei.

Reviewer 3 ·

Basic reporting

This study is well written. The English is clear and easy to follow. There is enough background information and references for the previous understanding of this threatened species. The figures and tables are clear. The figures are very informative and well explained.

Experimental design

I believe this study is well suited for the journal's interest. The questions of the study are important for the conservation of this species and answered by a thorough research combining tracking and ground verification.

Validity of the findings

The statistics is used right and supports the findings.

Here are several improvements and clarification that the author needs to address.

1. The transmitting cycles were different in three years. How did you reconciles the differences and turn into a comparable format?

2. The 114E is used as the midline and reference line. However, some stopover sites are also located before the birds getting there. What is the justification of still comparing the date crossing 114E if stopover sites are important part of migration and at the same time they are around 114E?

3. Line 179-181, add statistics and p value to the main text.

4. Line 221, please mention which month the maximum of birds were observed to make it clear.

5. results for pre-breeding dispersal. How many individuals displayed this pattern? Clarify the number of individuals for each site when counting the number of trips and the accumulative days.

Additional comments

The study is well designed and stated. As weather is an important part to explain the variation, it will be better to find weather data to support your speculation. This will make the argument stronger.

---

## Round 0.2 · accepted · Accept

Thanks for your prompt revision. I think you have addressed all the reviewers' comments clearly. Please remember that PeerJ does not copy edit the manuscript. For non-English speakers, I recommend checking the text with a programme such as Grammarly. I use it myself, incidentally. Next time I'm in China in the winter I'll have to look for this most interesting gull — it's not one I've seen!